Deep water vetulicolians from the lower Cambrian of China

Ma Shuhan 1 2
http://orcid.org/0000-0001-8032-4272 Kimmig Julien 3 4 julien.kimmig@smnk.de
http://orcid.org/0000-0003-4726-0355 Schiffbauer James D. 5 6
Li Ruibo 1 2
Peng Shanchi 7
Yang Xianfeng 1 2 yangxf@ynu.edu.cn
1 Yunnan Key Laboratory for Palaeobiology, Institute of Palaeontology, Yunnan University , Kunming , China
2 MEC International Joint Laboratory for Palaeobiology and Palaeoenvironment, Yunnan University , Kunming , China
3 The Harold Hamm School of Geology and Geological Engineering, University of North Dakota , Grand Forks, North Dakota , United States
4 Geosciences, State Museum of Natural History Karlsruhe , Karlsruhe , Germany
5 Department of Geological Sciences, University of Missouri , Columbia, Missouri , United States
6 X-ray Microanalysis Laboratory, University of Missouri-Columbia , Columbia, Missouri , United States
7 State Key Laboratory of Palaeobiology and Stratigraphy, Chinese Academy of Sciences , Nanjing , China
Corte Guilherme
Electronic publication date: 2025 Jan 22
Publication date: 2025
Volume: 13
Electronic Location ID: e18864
Received 2024 Oct 3; Accepted 2024 Dec 23
Copyright: © 2025 Ma et al.
Copyright year: 2025
Copyright holder: Ma et al.
License: This is an open access article distributed under the terms of the Creative Commons Attribution License, which permits unrestricted use, distribution, reproduction and adaptation in any medium and for any purpose provided that it is properly attributed. For attribution, the original author(s), title, publication source (PeerJ) and either DOI or URL of the article must be cited.
License URL: https://creativecommons.org/licenses/by/4.0/

Keywords: Vetulicolia, Niche, Exceptional preservation, Palaeoecology, Palaeogeography, Balang Formation, Chengjiang Biota, Burgess Shale

Funding: National Natural Science Foundation of China 42162001 Yunnan Provincial Research 2024y011 NSF CAREER 1652351 University of Missouri Marie M. and Harry L. Smith Endowment Financial support was provided by the National Natural Science Foundation of China (Grant No. 42162001) to Xianfeng Yang and the Yunnan Provincial Research Grant (Grant No. 2024y011) to Shuhan Ma. James D Schiffbauer is supported by NSF CAREER 1652351 and the University of Missouri Marie M. and Harry L. Smith Endowment. The funders had no role in study design, data collection and analysis, decision to publish, or preparation of the manuscript.

==============================
Vetulicolians are an enigmatic phylum of extinct Cambrian marine invertebrates. They are particularly diverse in the Chengjiang Biota of China, but representatives have been recovered from other Fossil-Lagerstätten (Cambrian Stage 3-Drumian). These organisms are characterized by a bipartite body, which is split into an anterior section and a posterior segmented section connected by a narrow constriction. Here we report new material of the genus Pomatrum from the Cambrian Balang Biota (Series 2, Stage 4) of Hunan, southern China. This is the first discovery of this vetulicolian outside of the Chengjiang Biota (Series 2, Stage 3) and the first report of vetulicolians from the Balang Biota. This finding not only suggests that this group had a wider spatial and temporal distribution than previously known, but also adds information to the overall biodiversity of the Balang Biota—one of the most important Stage 4 fossil deposits known from Gondwana.

Introduction

Vetulicolians are a problematic group of extinct marine invertebrates found in Cambrian Fossil-Lagerstätten (sensu Kimmig & Schiffbauer, 2024), with 16 species amongst ten genera that have been described so far, with more awaiting description (Table 1) (Walcott, 1911; Hou, 1987; Chen & Zhou, 1997; Luo et al., 1999; Shu et al., 2001; Chen et al., 2003; Briggs et al., 2005; Luo et al., 2005; Shu, 2005; Aldridge et al., 2007; Yang et al., 2010; Vinther, Smith & Harper, 2011; García-Bellido et al., 2014; Conway Morris et al., 2015; Fu et al., 2019; Kimmig, Leibach & Lieberman, 2020). Vetulicolians have a peculiar body structure, divided into an anterior and posterior body. The bilaterally symmetrical anterior body is usually weakly divided by five orderly lines and preserves five lateral pouches on each side (Ou et al., 2012; Li et al., 2018). The bilaterally symmetrical posterior body is generally paddle- or broad-leaf-shaped, and clearly divided into seven or more segments. It houses the gut and has a terminal anus. Additionally, some species preserve an enigmatic structure in the ventral position of their posterior ‘tail’. Though this feature has been suggested to be a likely internal organ, its function is currently unknown with inferences related to reproduction, excretion, and digestion (Yang et al., 2023b).

Table 1 Vetulicolian species in time and space.

Family	Species	Occurrence	Cambrian stage	References	
Vetulicolidae (Hou & Bergström, 1997)	Vetulicola cuneata Hou, 1987	Chengjiang Biota	Stage 3	Hou (1987), Aldridge et al. (2007)	
Vetulicola monile Aldridge et al., 2007	Chengjiang Biota	Stage 3	Aldridge et al. (2007)	
Vetulicola rectangulata Luo & Hu in Luo et al., 1999	Chengjiang Biota	Stage 3	Luo et al. (1999), Aldridge et al. (2007)	
Vetulicola gangtoucunensis Luo, Fu & Hu in Luo et al., 2005	Guanshan Biota	Stage 4	Luo et al. (2005), Hu et al. (2013)	
Vetulicola longbaoshanensis Yang et al., 2010	Guanshan Biota	Stage 4	Yang et al. (2010), Hu et al. (2013)	
Vetulicola sp.	Malong Biota	Stage 3	Luo et al. (2008)	
Vetulicola sp.	Xiaoshiba Biota	Stage 3	S. Ma, 2024, personal observations	
Vetulicola sp.	Shipai Biota	Stage 4	Zhang & Hua (2005)	
Vetulicola sp.	Jianhe Biota	Stage 4	Yang (2019)	
Vetulicola sp.	Guanshan Biota	Stage 4	Li et al. (2015)	
Beidazoon venustum Shu, 2005 (= Bullivetula variola Aldridge et al., 2007)	Chengjiang Biota	Stage 3	Shu (2005), Aldridge et al. (2007)	
Ooedigera peeli Vinther, Smith & Harper, 2011	Sirius Passet	Stage 3	Vinther, Smith & Harper (2011)	
Didazoonidae
(Shu et al., 2001)	Didazoon haoae Shu et al., 2001	Chengjiang Biota	Stage 3	Shu et al. (2001), Aldridge et al. (2007), Yang et al. (2023b)	
Pomatrum ventralis Luo & Hu in Luo et al., 1999 (= Xidazoon Shu et al., 1999)	Chengjiang Biota, Balang Biota	Stage 3–Stage 4	Luo et al. (1999), Shu et al. (1999), Aldridge et al. (2007), Yang et al. (2023b), this study	
Nesonektris aldridgei García-Bellido et al., 2014	Emu Bay Shale	Stage 3	García-Bellido et al. (2014)	
Yuyuanozoon magnificissimi Chen, Feng & Zhu in Chen et al., 2003	Chengjiang Biota	Stage 3	Chen et al. (2003), Aldridge et al. (2007), Li et al. (2018)	
Banffidae
(Caron, 2006)	Banffia constricta Walcott, 1911	Burgess shale	Wuliuan	Caron (2006), Aldridge et al. (2007)	
Banffia episoma Conway Morris & Selden in Conway Morris et al., 2015	Spence Shale	Wuliuan	Conway Morris et al. (2015), Kimmig et al. (2019)	
Banffia sp.	Qingjiang Biota	Stage 3	Fu et al. (2019)	
Heteromorphus confusus Chen & Zhou, 1997 (= Banffia confiusa Chen & Zhou, 1997)	Chengiiang Biota	Stage 3	Chen & Zhou (1997), Aldridge et al. (2007), Yang et al. (2021)	
Heteromorphus longicaudatus Luo & Hu, in Luo et al., 1999	Chengiiang Biota	Stage 3	Luo et al. (1999), Aldridge et al. (2007)	
Skeemella clavula Briggs et al., 2005	Marjum Formation,
Pierson Cove Formation	Drumian	Briggs et al. (2005), Aldridge et al. (2007), Kimmig, Leibach & Lieberman (2020)	
Indet.	new vetulicolian A	Qingjiang Biota	Stage 3	Fu et al. (2019)	
new vetulicolian B	Qingjiang Biota	Stage 3	Fu et al. (2019)	

The peculiar body shape of vetulicolians has led to continued conversation regarding their phylogenetic placement. Originally, vetulicolians were considered a type of large bivalved arthropod (e.g., Hou, 1987; Caron, 2001). However, the consistent absence of appendages in an increasing number of vetulicolian species has swayed most away from this position, instead resulting in propositions of several other affinities, including deuterostomes (e.g., Shu et al., 2001), stem-chordates (Gee, 2001), tunicates (e.g., Lacalli, 2002), or ecdysozoans (e.g., Briggs et al., 2005). The prevailing view is that vetulicolians are a monophyletic group within the deuterostomes, perhaps closest to free-swimming tunicates, but their precise position within the Deuterostomia is still debated (García-Bellido et al., 2014; Li, Liu & Ou, 2017; Li et al., 2018; Zhu et al., 2019; Shu & Han, 2020; Yang et al., 2023b). Indeed, the most recent phylogenetic analysis (Mussini et al., 2024) changes course again, suggesting a paraphyletic group amongst stem-chordates, with the Didazoonids placed most closely to Yunnanozoon. Another hypothesis would place vetulicolians within the stem-group protostomes under a scenario of deuterostome paraphyly (Kapli et al., 2021).

Vetulicolians exhibit a relatively wide distribution in space, as they are known from Cambrian Stage 3–Stage 4 of Gondwana and the Wuliuan-Drumian of Laurentia (Table 1). However, the majority of vetulicolians come from South China, and most species are restricted to the Chiungchussu Formation of South China (Table 1). This makes the recent discovery of Pomatrum cf. P. ventralis from the Balang Formation (Cambrian, Series 2, Stage 4), Xiangxi Region, Hunan, China, significant, as it represents the first appearance of the genus outside of the Chengjiang Biota (Series 2, Stage 3). Here we describe ten new specimens of Pomatrum, explore the distribution of the genus in time and space, and discuss potential adaptations of vetulicolians to deeper water environments.

Materials and Methods

The new specimens of Pomatrum cf. P. ventralis described here are reposited in the collections of the Yunnan Key Laboratory for Palaeobiology, Yunnan University, Kunming, China (YKLP), with specimen numbers YKLP 14,591–14,601. Specimens with the prefix Hz-f are reposited in the collections of the Yunnan Institute of Geological Sciences, Kunming, China.

Imaging

The specimens were photographed using a Canon EOS 5D digital SLR camera with a Canon 50 mm macro lens, cross-polarized lighting, and one-sided, low-angle lighting. Close-ups were captured using a Leica DFC 500 digital camera mounted on a Leica M205-C stereoscope. The specimens were submerged in water to increase contrast. The contrast, color space, and brightness were adjusted using Adobe Photoshop CC.

Terminology

The vetulicolian terminology in our descriptions broadly follows Li et al. (2018).

Geological setting

The Balang Formation is exposed in the eastern Guizhou and western Hunan Provinces of southern China (Zhou et al., 1979, 1980; Yin, 1996; Yang et al., 2023a). The thickness of the Balang Formation varies between 300 and 600 m in the Guizhou Province and is generally about 200 m in the Hunan Province. The specimens described herein come from an outcrop approximately 32 km south-west of Huayuan Town, Xiangxi Region, Hunan Province, South China (Fig. 1A). At this location, the Balang Formation conformably overlies the lower Cambrian Niutitang Formation and is in turn conformably overlain by the Chinghsutung Formation. The Balang Formation itself can be divided into a lower and an upper section. The lower part of the section contains abundant small planktonic trilobites such as Oryctocephalidae, reflecting a deeper water environment. The upper part of the section contains abundant benthic trilobites belonging to the Ptychopariida and Redlichiida (Yin, 1996).

Figure 1 Location and stratigraphy.

(A) Location of the studied section of the Balang Formation. It is located approximately 32 km south-west of Huayuan Town, Xiangxi Region, Hunan Province, South China. Lower Cambrian Outcrops in grey. Fossil side indicated by red star. (B) Generalized stratigraphy of the Balang Formation at this location. Modified from Yang et al. (2023a). (C) Palaeogeographical distribution of Pomatrum specimens during Cambrian Stages 3–4 and during the Wuliuan (map modified from Streng & Geyer (2019)). 1. Chengjiang Biota, Yunnan, China, Stage 3, 2. Balang Biota, Hunan, China, Stage 4.

The vetulicolian specimens here were collected from the dark grey calcareous mudstones of the lower part of the Balang Formation (Fig. 1B). The abundance of the trilobite Oryctocarella duyunensis in the same beds places these units within Cambrian Series 2, Stage 3–4 (Peng et al., 2017; National Commission on Stratigraphy of China, 2018; Zhao et al., 2019; Dai et al., 2021). Specimens from this location are preserved as carbonaceous compressions (Wen et al., 2019; Yang et al., 2023a).

Results

Systematic Paleontology

Phylum: Vetulicolia Shu et al., 2001

Class: Vetulicolida Chen & Zhou, 1997

Order: Vetulicolata Hou & Bergström, 1997

Family: Didazoonidae Shu & Han in Shu et al., 2001

Genus Pomatrum Luo & Hu in Luo et al., 1999

Xidazoon.–Shu, Conway Morris & Zhang, 1999: 747.

Remarks. We follow Chen et al. (2002) and Aldridge et al. (2007) in considering Xidazoon a junior synonym of Pomatrum.

Pomatrum cf. P. ventralis.–Luo & Hu in Luo et al., 1999

Pomatrum ventralis.–Luo and Hu, in Luo et al., 1999: 65–66, pl. 12, fig. 3.

Xidazoon stephanus.–Shu, Conway Morris & Zhang, in Shu et al., 1999: 747–749, figs. 1a–d, 2.

Xidazoon stephanus.–Shu et al., 2001: figs 2e–f, 3c–d.

Pomatrum ventralis.–Chen et al., 2002: pl. 12, fig. 4.

Xidazoon stephanus.–Shu, 2003: fig. 3b.

Pomatrum ventralis.–Chen, 2004: 309, 315, figs. 493–494, 501, 505.

Pomatrum ventralis.–Aldridge et al., 2007: 146–147, pl. 4, figs. 6–8, 10–12.

Xidazoon stephanus.–Yang et al., 2023b: 4, figs. 1E, H, 2, 3, S1–S7, S8C–F.

Holotype. Hz-f-7-390.

New Material. YKLP 14591-14592 (part and counterpart of a nearly complete laterally compressed specimen (Figs. 2A–2F). YKLP 14593-14601, nine laterally compressed incomplete specimens on a single slab (Figs. 3A, 3B).

Figure 2 Nearly complete specimen of Pomatrum cf. P. ventralis Luo & Hu in Luo et al. (1999) from the Cambrian Stage 4 Balang Formation of China.

(A) YKLP 14591, part, a relatively complete specimen of preserving the anterior and posterior body, lateral pouch, lateral groove and alimentary canal. (B) Explanatory drawing of (A). (C) YKLP 14592 counterpart, preserving the first three subdivisions of the anterior section, two lateral pouches, lateral groove and marginal zone. (D) Close-up of the anterior opening, indicated by arrows. (E) Close-up of the lateral pouches and lateral groove from (C). (F) Close-up of the alimentary canal and the enigmatic sub-rounded structure of the posterior section, indicated by arrows. Abbreviations: Ac, Alimentary canal; As, Anterior section; Ir, Inner region; Lg, Lateral groove; Lp, lateral pouch; Mz, Marginal zone; Oo, Oral opening; Or, Outer region; Ps, Posterior section; S, Segment.

Figure 3 Clustered partial specimens of Pomatrum cf. P. ventralis Luo & Hu in Luo et al. (1999) from the Cambrian Stage 4 Balang Formation of China.

(A) YKLP 14593-14561, six anterior specimens, one posterior specimen and two unidentifiable fragments of Pomatrum cf. P. ventralis. (B) Interpretative drawing of (A).

Provenance. Balang Formation, lower Cambrian (Series 2, Stage 3-4), Oryctocarella duyunensis biozone, Mozi Village, Paiwu Township, approximately 32 km south-west of Huayuan Town, Xiangxi Region, Hunan Province, China. Yu’anshan Member, Chiungchussu Formation, lower Cambrian (Series 2, Stage 3), Wutingaspis–Eoredlichia biozone, Yunnan, China.

Emended diagnosis. Vetulicolian with a weakly sclerotized, bipartite body. Anterior body with smooth surface, ovoid with weak segmentation, most prominent anteriorly, with circular anterior opening surrounded by prominent plates divided into inner and outer regions. Either side of anterior section with a longitudinal series of five oval-shaped lateral pouches, each covered by a hood-like structure. Posterior ventral margin of anterior section with a narrow finlike structure. Posterior section, laterally flattened, tapering anteriorly and posteriorly, with seven segments. Anus located near the center of the final segment of the posterior section, followed by a terminal notch. A sub-rounded, striated structure located at or between the third and fourth tail segments (modified from Aldridge et al., 2007 and Yang et al., 2023b).

Description. Best-preserved specimen includes part (YKLP 14591) and counterpart (YKLP 14592) (Fig. 2). Specimen bipartite, divided into anterior and posterior sections. Anterior ovoid, missing part of the dorsal margin. Weakly subdivided by ordering lines, only the anterior most subdivisions are visible (Fig. 2A). Surface smooth. Anterior section is 50 mm long and 25 mm wide at its widest point. First subdivision preserves a circular anterior opening. Inner anterior disc surrounds the opening, visible as a darkened disc, displays some very fine lines radiating from the opening (Fig. 2D). Outer anterior disc region less well preserved, can be distinguished from the inner disc by a change in color (Fig. 2D). Lateral grove connects the opening and four ovoid lateral pouches; the fifth lateral pouch is not preserved. Anterior subdivisions preserves four ordering lines (Fig. 2B) only one crosses the whole body. Other ordering lines incomplete or missing. Diameter of the lateral pouches ranges from 2 to 5 mm.

The posterior section of YKLP 14591 appears to be detached from the anterior section. Anterior section 30 mm long and 15 mm wide at its widest point. Divided into seven segments (Fig. 2A). Alimentary canal located along the central axis, mostly complete, however, the sedimentary infill is intermittent, but locally present (Fig. 2F). Anus and notch not preserved. In segment 4, structure resembling the enigmatic sub-rounded structure preserved (Fig. 2F) described by Yang et al. (2023b).

Counterpart (YKLP 14592) only preserves the first three segments of anterior section, oral opening, and two lateral pouches (maximum diameter 5 mm) (Figs. 2C, 2E).

In addition to the mostly complete specimen, nine (YKLP14593–YKLP14601) incomplete specimens are preserved on a single slab (Fig. 3). Six anterior sections, based on preserved lateral pouches. Their length varies between 22 and 34 mm, and the maximum width ranges from 18 to 25 mm. One specimen (Fig. 3B, #8) represents a posterior body. It preserves five segments, and most of the alimentary canal. This specimen is 31 mm long and 21 mm at its widest point. The other two specimens on this slab are not preserved well enough to identify them as anterior or posterior.

Remarks. The Balang Formation vetulicolians are tentatively assigned to Pomatrum ventralis based on the following structures they have in common with the specimens from the Chengjiang Biota (Luo & Hu in Luo et al., 1999; Shu et al., 1999; Aldridge et al., 2007; Yang et al., 2023b): a bipartite ovoid anterior section weakly subdivided by ordering lines, a circular anterior opening divided into inner and outer regions, the presence of oval-shaped lateral pouches connected by a lateral groove, a posterior section that is laterally flattened, tapering anteriorly and posteriorly, preserves seven segments, a medial alementary canal and includes the presence of a sub-rounded structure located at the fourth tail segment. The tentative assignment is due to the incomplete preservation of all ten specimens. However, none of the morphological features suggest that the specimens might belong to a different species or even genus.

Discussion

The oldest, and until now only known, occurrences of Pomatrum ventralis are from the lower Cambrian (Series 2, Stage 3) Chengjiang Biota of China, where hundreds of specimens have been recovered (Shu et al., 1999; Aldridge et al., 2007; Hou et al., 2017; Yang et al., 2023b). The specimens reported herein from the Balang Formation in Xiangxi Region, Hunan Province, are younger (Series 2, Stage 3–4), and preserved in dark grey calcareous mudstones. The limited distribution of the species in time and space suggests that this species might have been restricted to the lower Cambrian of Gondwana, however there are more Konservat-Lagerstätten (sensu Kimmig & Schiffbauer, 2024) in this time interval in Gondwana than in Laurentia, which might lead to a biased view (sensu Whitaker & Kimmig, 2020) of its distribution.

This discovery of Pomatrum in deeper water sediments implores consideration of how broad the niche of Pomatrum, and vetulicolians in general, might have been as early as the lower Cambrian. Based on published data, the majority of vetulicolian species, including Pomatrum, are found in the Chengjiang Biota (Shu, 2005; Aldridge et al., 2007). This suggests that they likely preferred shallower, better oxygenated waters in the photic zone, as the Chengjiang Biota has been considered to record faunas from deltaic to inner shelf environments (Saleh et al., 2022). While few in number, the majority of the Balang Formation specimens (Figs. 2, 3) reported here are from a single slab (Fig. 3) and aligned in one direction, which suggests that the specimens were likely transported postmortem by currents, or represent a small die-off with clustered animals open to decomposition on the seafloor before burial. The presence of complete and exquisitely preserved Herpetogaster collinsi specimens in the same beds as the Pomatrum specimens (Yang et al., 2023a) suggest that transportation was probably minimal, and that the dismemberment of the Pomatrum specimens was more likely due to decomposition. In turn, this suggests that Pomatrum was living in deeper water environments.

While finding vetulicolians in deeper water environments in the lower Cambrian is unusual compared to other known Chinese specimens (Fig. 4), the Laurentian species of the Wuliuan and Drumian are all found in deeper water deposits (Briggs et al., 2005; Caron, 2006; Aldridge et al., 2007; Conway Morris et al., 2015; Kimmig, Leibach & Lieberman, 2020). This may imply an overall niche evolution for vetulicolians from shallow water environments in Cambrian Stage 3 to deeper water environments by Stage 4 (e.g., Hou, 1987; Luo et al., 2005; Zhang & Hua, 2005; Luo et al., 2008; Yang et al., 2010; Li et al., 2015; Li, Liu & Ou, 2017; Chen et al., 2019; Fu et al., 2019; Yang, 2019). If the signal is true and not an exposure or a taphonomic bias in different time slices-and there are currently no shallow water Konservat-Lagerstätten that offer a palaeoenvironmental counterpoint to the Chengjiang Biota known from the Wuliuan or Drumian (Muscente et al., 2017), a question that arises is what forces drove vetulicolians into deeper water environments? Increasing oxygenation of deeper waters and/or changes to resource availability may have been contributing factors (e.g., Stockey et al., 2024). A general trend to invade deeper water environments for Cambrian organisms, potentially linked to environmental pressures or biotic refuges, has been suggested (e.g., Conway Morris, 1989 and references therein). However, this hypothesis will only be testable through shallow water Konservat-Lagerstätten from the middle and late Cambrian.

Figure 4 Distribution of vetulicolians in South China.

(A) Paleogeographical distribution of vetulicolians in South China (Cambrian, Series 2, Stages 3–4). (B) Depositional environment of Pomatrum in South China (Cambrian, Series 2, Stages 3–4). Abbreviations: Bl, Balang Formation; Cj, Chengjiang Biota; Gs, Guanshan Biota; Ml, Malong Biota; Qj, Qingjiang Biota; Sp, Shipai Biota; Tst, Tsinghsutung Formation (Jianhe Biota); Xsb, Xiaoshiba Biota.

The Balang Formation preserves a relatively diverse fossil assemblage (e.g., Liu & Lei, 2013; Yang et al., 2023a) and the limited amount of taxonomic work it has received suggests, that the diversity is significantly higher than currently known. The lower Balang Formation deposit likely represented a depositional environment similar to the deeper water deposits of the Wulian Spence Shale in the Wellsville Mountains (Kimmig et al., 2019) or the Burgess Shale (Nanglu, Caron & Gaines, 2020). This is supported by the presence of several taxonomic groups that are usually associated with deeper water environments, like planktonic oryctocephalid trilobites (Yin, 1996) and Herpetogaster (Yang et al., 2023a), as well as the dark grey calcareous mudstones that preserve the biota.With each new discovery, it is also becoming more apparent that the taxonomic composition and species diversity of the Balang Biota are a unique window into the deeper-water Konservat-Lagerstätten of Gondwana. This deposit is currently also the most comparable Konservat-Lagerstätte to the Wuliuan and Drumian ones of Laurentia, not only in biologic diversity and depositional environment, but also in the type of preservation of the fossils (Wen et al., 2019; Yang et al., 2023a). More research on this and similar deposits will provide a more complete picture of the biodiversity and paleoecology of the lower Cambrian deeper water environments and will contribute to the understanding of the early evolution of these habitats.

Conclusions

The Balang specimens of Pomatrum cf. P. ventralis represent the first report of this species from outside the Chengjiang Biota. The detailed preservation of the specimens, which includes the characteristic lateral pouches and the circular oral opening surrounded by two circular discs, makes an assignment to the genus unquestionable. This extends the range of the species into Cambrian Series 4. Additionally, the new specimens provide evidence for a wider ecological niche in early vetulicolians, exploring deeper water environments as early as late Cambrian Stage 4.

This article is a contribution to IGCP668, Equatorial Gondwanan History and Early Palaeozoic Evolutionary Dynamics. We thank the reviewers and the editor for their constructive suggestions.

Additional Information and Declarations

Competing Interests

Author Contributions

Data Availability

The authors declare that they have no competing interests.

Shuhan Ma conceived and designed the experiments, performed the experiments, analyzed the data, prepared figures and/or tables, authored or reviewed drafts of the article, and approved the final draft.

Julien Kimmig analyzed the data, prepared figures and/or tables, authored or reviewed drafts of the article, and approved the final draft.

James D. Schiffbauer analyzed the data, authored or reviewed drafts of the article, and approved the final draft.

Ruibo Li analyzed the data, authored or reviewed drafts of the article, and approved the final draft.

Shanchi Peng analyzed the data, authored or reviewed drafts of the article, and approved the final draft.

Xianfeng Yang conceived and designed the experiments, analyzed the data, prepared figures and/or tables, authored or reviewed drafts of the article, and approved the final draft.

The following information was supplied regarding data availability:

The specimen data is available in Table 1.

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
