# Peer review of "Deep water vetulicolians from the lower Cambrian of China"

_PeerJ, doi:10.7717/peerj.18864_

## Round 0.1 · original submission · Major Revisions

Dear Dr. Kimmig,

Your paper has been reviewed by two experts in the field. They agree that your research was well-executed, and your manuscript provides relevant information on the vetulicolian fossils from the Cambrian. However, they also provided important suggestions that I hope you address. After you revise the manuscript following the reviewer's suggestions, I will be pleased to reconsider the manuscript for publication in PeerJ. Please make sure to acknowledge the reviewer's valuable contributions to the revised version.

Reviewer 1 ·

Basic reporting

This study presents new material of the genus Pomatrum from the Cambrian Balang Biota of China. The introduction is well-structured and provides essential context. The figures are relevant, and the specimens have sufficient characteristics for identification. The discovery is noteworthy and deserves to be published. Prior to its publication, it needs to be put more in context with existing literature, and certain aspects of the ms need review and correction.

Experimental design

This study is well-aligned with the journal's scope, and the questions are well-defined filling a knowledge gap regarding the presence of the vetulicolian Pomatrum outside of the Chengjiang Biota. The methodologies and observations conducted are both professional and ethically sound. The methods described are sufficient for replication.

Validity of the findings

All data are provided, with conclusions clearly articulated and connected to the research question and supporting results.

Additional comments

Line 29-30- I would discourage the use of “head shield-like region” and “segmented tail-like region”, given that it may be confusing to some readers, who may read them as an arthropod terminology (or if you agree with this affinity, further explanation is needed). Please revise them and ensure the terminology are consistent throughout the ms.

Line 68- Please explain why the authors propose that vetulicolians exhibit a relatively wide distribution in time, despite currently being known only from the Cambrian period.

Line 80- Material should include all specimens (YKLP 145591-14592, 14593-14601, and specimens with the prefix Hz-f). Replace “specimen” with “specimens”; replace “is” with “are”

Line 91- Use the full stop (.) at the end of the sentence.

Line 99- Should be “Huayuan Town” “South China”. Proper nouns should always be capitalized. Please check the entire ms and make adjustments accordingly.

Line 145- Replace “partial” with “incomplete”

Line 163-189- Description should be articulated more concisely in telegraphic style. Please revise this text accordingly.

Line 216- Multiple specimens are facing in the same direction. Why do you believe it was a post-mortem burial rather than in-situ?

Line 294- References' format needs to be revised (e.g. author names, Hou XG, Hou X., Hou X-G, Shu D., Shu DG).

Line 512- Chengjiang biota
Line 531- Chengjiang Biota
Please keep the capitalization of the same noun consistent throughout the entire ms.

Line 531- 532 (Cambrian, Series 2, Stages 3-4,) Remove the last “,”

Table 1. There are several papers published in the past years that are dismissed in the table and deserve to be included. For example,
Vetulicola sp. from Guanshan biota (Li et al., 2015 in Palaeoworld)
Didazoon haoae (Yang et al., 2023 in Papers in Palaeontology)
Pomatrum ventralis (Shu et al., 1999 in Nature)
Yuyuanozoon magnificissimi (Li et al., 2018 in Journal of Paleontology)

For publication, I would recommend the revised ms provide, if possible, a clearer morphological comparison of Pomatrum from Chengjiang and Balang, as well as the evolutionary trend.

Reviewer 2 ·

Basic reporting

Ma et al. present a set of new vetulicolian fossils from the Cambrian Series 2 Balang biota, recording the first early Cambrian occurrence of vetulicolians in this setting and in deeper-water biotas more generally. They attribute the new fossils to the genus Pomatrum, previously reported from the Chengjiang biota. The authors proceed to discuss how the new fossils expand the known palaeogeographical and palaeoenvironmental range of vetulicolians.

The text is well structured, and the figures of sufficient quality. The strength of the evidence is mostly adequate to support the conclusions presented in the manuscript. However, I would recommend the inclusion of more precise background and contextual information in relation to the palaeoenvironmental hypotheses presented in the Discussion.

Experimental design

no comment

Validity of the findings

The hypotheses presented in the Discussion are speculative in places. This is not a problem in itself, since it is adequately acknowledged. However, the readers would benefit from more precise contextual data to evaluate the relative strength of alternative hypotheses, or even assess whether they are testable at all.

Additional comments

Specific comments:

44: “with 16 species amongst ten genera [that] have been described so far, with more awaiting description” (Table 1).
49: “The bilaterally symmetrical 50 anterior body” – the posterior body is bilaterally symmetrical, too.
50: is usually weakly divided into six segments – there are clearly annulations on the anterior body, but is “segmentation” the right word here? It may give the misleading impression that the same kind of metamerism is present in the anterior and posterior bodies.
62: “The current consensus is that vetulicolians are a monophyletic group” – you then proceed to mention how the most recent study suggests paraphyly (so not a consensus – perhaps a most common/prevailing view).
63: “their precise position within the 64 Deuterostomia is still debated” – it may be worth referencing Kapli et al. 2021 – Lack of support for Deuterostomia prompts reinterpretation of the first Bilateria – to also acknowledge the hypothesis of vetulicolians as stem-group protostomes under a scenario of deuterostome paraphyly.
67: “the Didazoonids placed most closely to Yuyuanozoon and Yunnanozoon.” Suggest changing to "the Didazoonids placed most closely to Yunnanozoon" (Yuyuanozoon is considered part of Didazoonidae by Mussini et al. and previous authors).
159: “near the center of” – medially?
178: “the 178 sedimentary infill is missing (Fig. 2F).” it looks like it’s intermittent, but locally present.
179: “In segment 4, there is a structure resembling the enigmatic sub-rounded structure (Fig. 2F)” This feature should be arrowed or otherwise highlighted on the figure, particularly since it’s discussed in the Remarks to strengthen attribution to Pomatrum ventralis.
205: “this genus” – you refer to Pomatrum ventralis here – so you seem to attribute the new fossils down to species level. More broadly, I think that given the incomplete preservation of the new Balang fossils it would be safest to attribute them simply to Pomatrum, rather than to its type species ventralis. While the evidence for genus-level attribution is robust (particularly given the two diagnostic oral “circlets” visible in the fossils), based on the material at hand I would be more cautious to make statements about the Chengjiang and Balang material representing the same species.
212: “This suggests that they likely preferred shallower, higher oxygenated waters in the photic zone, as the Chengjiang biota has been considered a deltaic to inner shelf environment”. Substitute “higher” for “better”. It may be worth clarifying that the Chengjiang Lagerstätte was not deposited in oxygenated waters itself, since it records mostly transported fauna (in the words of Saleh et al., “The lack of oxygen in the prodelta suggests that most animals must have been living in shallower oxygenated waters and were transported by flows from the delta front to the prodelta where preservation occurred.”). Or, the phrasing could be changed to something like “the Chengjiang biota has been considered to record faunas from deltaic to inner shelf environments” to avoid confusion between the animals’ original habitats and their burial environments.
224: “the Laurentian species of the Wuliuan and Drumian are all found in deeper water deposits” and 230: “If the signal is true and not an exposure or a taphonomic bias in different time slices”. Can the authors give examples of Wuliuan and Drumian Lagerstätten that are not from deeper-water deposits? To my mind, the Spence Shale is the only one that (in places) may come close to this description among truly “exceptional” localities; I cannot think of any Wuliuan or Drumian biotas that offer a palaeoenvironmental counterpoint to the Chengjiang. If so, before considering what forces drove vetulicolians into deeper water environments it should be at least made clearer that “exposure or a taphonomic [or rather, palaeoenvironmental] bias in different time slices” is very real in this case: we lack a temporally commensurate record of shallow versus deeper-water middle Cambrian Lagerstätten.
237: “likely represented an ecosystem similar to the Wulian Spence Shale (Kimmig et al., 2019) or 238 Burgess Shale (Nanglu et al., 2020).” It is not clear to me what is being argued here: similar in terms of taxonomic composition, species richness/diversity, functional guilds, environment, all of the above? The Spence Shale and the Burgess Shale are not equivalent palaeoenvironmentally – the Spence captures a broader spectrum of shallow-water carbonate to deep-shelf dark shale deposits with greater variability in bioturbation and oxygenation than the Burgess, so it may be worth clarifying what the “ecosystem” comparison is addressed at.

Overall, this is an interesting paper and I look forward to seeing it published.

---

## Round 0.2 · Minor Revisions

Dear Dr. Kimmig,

Thank you for reviewing your manuscript following the reviewer's recommendations. Both reviewers agree that your manuscript is almost ready for publication. Before the final acceptance, I kindly ask you to consider the few suggestions provided by reviewer 2. Then, I will be pleased to accept the manuscript for publication in PeerJ.

Best regards,

Guilherme

Reviewer 1 ·

Basic reporting

The authors have responded promptly and comprehensively to the reviewers' comments, and I believe that the manuscript is ready for publication.

Experimental design

No further comments.

Validity of the findings

No further comments.

Additional comments

No further comments.

Reviewer 2 ·

Basic reporting

I would like to thank the authors for taking my suggestions into account. I have no substantial points left to make - only some minor recommended edits to the text:

line 271 - "The detailed preservation of the specimens, which includes the characteristic lateral pouches and the circular oral opening surrounded by two circular discs, makes an assignment to the species unquestionable." The authors attribute the fossils to Pomatrum cf. ventralis, and based on the wider discussion it seems that it's the genus rather than species that is unquestionable.

line 260 - "an unique window" - should be "a unique window"

line 251 - "the relative little amount of taxonomic work" - should be "limited amount"

line 245 - "A general trend to invade deeper water environments for Cambrian organisms has been suggested (e.g., Conway Morris, 1989 and references therein), and was linked to environmental pressures or possible refuge, this hypothesis will however only be testable if shallow water Konservat-Lagerstätten from the middle and late Cambrian are discovered. " - Better structured along the lines of "A general trend to invade deeper water environments for Cambrian organisms, potentially linked to environmental pressures or biotic refuges, has been suggested (e.g., Conway Morris, 1989 and references therein). However, this hypothesis will only be testable through shallow water Konservat-Lagerstätten from the middle and late Cambrian."

line 240 - "If the signal is true and not an exposure or a taphonomic bias in different time slices, there are currently no shallow water Konservat-Lagerstätten that offer a palaeoenvironmental counterpoint to the Chengjiang Biota known from the Wuliuan or Drumian (Muscente et al., 2017), a question that arises is what forces drove vetulicolians into deeper water environments?
- may be better restructured into ""If the signal is true and not an exposure or a taphonomic bias in different time slices - and there are currently no shallow water Konservat-Lagerstätten that offer a palaeoenvironmental counterpoint to the Chengjiang Biota known from the Wuliuan or Drumian (Muscente et al., 2017) - a question that arises is what forces drove vetulicolians into deeper water environments?

line 219 - Pomatrumin shoudl be Pomatrum in

line 160 - for clarity, change Vetulicolid to vetulicolian (Pomatrum is a didazoonid) or use formal class name Vetulicolida

Experimental design

no comment

Validity of the findings

no comment

Additional comments

no comment

---

## Round 0.3 · accepted · Accept

Dear Dr. Kimmig,

It is my pleasure to inform you that your manuscript entitled "Deep water vetulicolians from the lower Cambrian of China" has been accepted for publication in PeerJ.

Thank you for sharing your time and expertise with the PeerJ community and beyond.

Best regards,

Guilherme Corte